# Physiological and Pathophysiological Effects of C-Type Natriuretic Peptide on the Heart

**DOI:** 10.3390/biology11060911

**Published:** 2022-06-14

**Authors:** Akihiro Yasoda

**Affiliations:** National Hospital Organization Kyoto Medical Center, 1-1, Mukaihata-cho, Fukakusa, Fushimi-ku, Kyoto 612-8555, Japan; ayasoda@kuhp.kyoto-u.ac.jp

**Keywords:** CNP, NPR-B, osteocrin

## Abstract

**Simple Summary:**

C-type natriuretic peptide (CNP) is the third member of the natriuretic peptide family. Unlike atrial natriuretic peptide (ANP) and brain natriuretic peptide (BNP), CNP was not previously regarded as an important cardiac modulator. However, recent studies have revealed the physiological and pathophysiological importance of CNP in the heart; in concert with its cognate natriuretic peptide receptor-B (NPR-B), CNP has come to be regarded as the major heart-protective natriuretic peptide in the failed heart. In this review, I introduce the history of research on CNP in the cardiac field.

**Abstract:**

C-type natriuretic peptide (CNP) is the third member of the natriuretic peptide family. Unlike other members, i.e., atrial natriuretic peptide (ANP) and brain natriuretic peptide (BNP), which are cardiac hormones secreted from the atrium and ventricle of the heart, respectively, CNP is regarded as an autocrine/paracrine regulator with broad expression in the body. Because of its low expression levels compared to ANP and BNP, early studies failed to show its existence and role in the heart. However, recent studies have revealed the physiological and pathophysiological importance of CNP in the heart; in concert with the distribution of its specific natriuretic peptide receptor-B (NPR-B), CNP has come to be regarded as the major heart-protective natriuretic peptide in the failed heart. NPR-B generates intracellular cyclic guanosine 3′,5′-monophosphate (cGMP) upon CNP binding, followed by various molecular effects including the activation of cGMP-dependent protein kinases, which generates diverse cytoprotective actions in cardiomyocytes, as well as in cardiac fibroblasts. CNP exerts negative inotropic and positive lusitropic responses in both normal and failing heart models. Furthermore, osteocrin, the intrinsic and specific ligand for the clearance receptor for natriuretic peptides, can augment the effects of CNP and may supply a novel therapeutic strategy for cardiac protection.

## 1. Introduction

C-type natriuretic peptide (CNP) was extracted from porcine brain in 1990 for the first time [1] and then cloned in pigs [2], as well as in rats [3] and humans [4]. CNP is the third member of the natriuretic peptide family, and, along with the other two natriuretic peptide family members, i.e., atrial natriuretic peptide (ANP) and brain natriuretic peptide (BNP), it shares a similar ring structure of 17 amino acids with the essential residues needed to exert their biological actions through binding to their biological active receptors [1] (Figure 1A). CNP acts as an intrinsic and bioactive peptide ligand through the binding to its specific membrane guanylyl cyclase receptor [5], natriuretic peptide receptor-B (NPR-B) [6], whereas ANP and BNP are selective ligands for the other receptor membrane guanylyl cyclase, natriuretic peptide receptor-A (NPR-A) [7]; these NPRs exert their biological action through the generation of the second messenger, cyclic guanosine 3′,5′-monophosphate (cGMP) from GTP upon ligand binding [8]. Accordingly, they are also referred to as guanylyl cyclase-A (GC-A) for NPR-A and guanylyl cyclase-B (GC-B) for NPR-B (Figure 1B).

Although ANP and BNP act as cardiac hormones secreted from the atrium and ventricle of the heart, respectively [9,10,11], CNP was shown to be expressed ubiquitously in the body [12]: CNP and its specific receptor, NPR-B [13], are expressed in the central nervous system including the brain [14], hypothalamus [15,16], and pituitary gland [14,17], in the circulatory system including the blood vessels [18] and heart, in the reproductive system [19,20], and in the skeletal system including the growth plate cartilage [21,22,23,24]. Together with the fact that CNP has considerably low blood concentrations in mammals [25,26,27], CNP is regarded as an autocrine/paracrine regulator, not an endocrine hormone [28].

For a few decades, the physiological and pathophysiological roles of ANP and BNP were intensively investigated, and these cardiac hormones were shown to be engaged in the protection of circulatory homeostasis, including the exertion of their cardioprotective effects. These discoveries have assisted in establishing the clinical implications of ANP and BNP as biomarkers that detect cardiac disease including heart failure and cardiac hypertrophy, and ANP and BNP themselves and molecules relevant to them have been implicated in the development of therapeutic agents for heart failure or related diseases [29,30].

On the other hand, along with the notion that CNP is a ubiquitously expressed autocrine/paracrine factor, research pursuing the physiological and pathophysiological roles of CNP in the body and their transition to clinical use has diverged into a wide variety of organs. The recent and most prominent studies were performed in the skeletal system [31,32]. Nevertheless, several groups have persisted with working to resolve the roles of CNP in the heart [33]. In this review, after a brief general description of CNP, I introduce studies of CNP in the heart.

## 2. General Features of CNP

### 2.1. Generation of CNP

The gene encoding CNP, *NPPC*, is located on the second chromosome, 2q37.1, in humans. CNP is first produced as the pre-pro-peptide of 126 amino acids, which is subsequently cleaved into proCNP with 103 amino acids by the endoprotease furin [34]. ProCNP is further cleaved by furin into biologically active and mature CNP with 53 amino acids (CNP-53) and the presumably bio-inactive N-terminal product amino-terminal proCNP with 50 amino acids (NTproCNP). A New Zealand group has been reporting on the significance of NTproCNP as a clinical biomarker for various physiological and pathophysiological conditions including issues concerning skeletal growth in humans and other experimental animal models [35]. In some cases, CNP-53 is further broken down into a variant with 22 amino acids (CNP-22), whose biological activity is thought to be equal to that of CNP-53, by an unknown enzyme [36] (Figure 1A).

As for the regulation of CNP production, cytokines (transforming growth factor β, tumor necrosis factor α, and interleukin 1β), bacterial endotoxin lipopolysaccharides [28], transcription factors (kruppel-like factor 2) [37], hypoxia [38], and shear stress [39] were reported to stimulate the production of CNP in blood vessels in in vivo and ex vivo experiments. Furthermore, molecules participating in Wnt signaling reportedly stimulated CNP production in the kidney [40], while transforming growth factor β stimulated CNP production in chondrocytes [41] and osteoblasts [21] in in vitro experiments. In the ovaries of mice, excess human chorionic gonadotropin treatment reportedly exhibited a marked decrease in CNP expression in granulosa cells of the preovulatory follicles [20].

### 2.2. Receptors for CNP and Their Downstream Signaling

As mentioned above, the biologically active receptor for CNP is NPR-B. NPR-B is a membranous receptor guanylyl cyclase that produces cGMP from GTP through selective ligand CNP binding. The downstream pathways of this CNP/NPR-B/cGMP signaling cascade include the pathways through cGMP-dependent protein kinases (abbreviated as cGKs or PKGs), cGMP-dependent phosphodiesterases (PDEs), and cGMP-gated ion channels, all of which cause a broad variety of physiological responses. Among these pathways, that through cGKs is regarded as the most important one; cGKs phosphorylate various downstream target proteins. One subtype of cGKs, cGKII, is reported to play an important role in skeletal growth as the downstream mediator of the CNP/NPR-B/cGMP pathway [42,43,44,45]. On the other hand, cGMP signaling is abrogated by cGMP hydrolysis via PDEs and cGMP export via multidrug resistance proteins.

Furthermore, there exists another receptor for CNP, named NPR-C, which was initially reported to be engaged in the clearance of ligands [46]. The catalytic effect of NPR-C is discussed in the next section. After the discovery of its clearance action, it was revealed that NPR-C contains Gi-binding domains in its intracellular C-terminal region, which induce the inhibition of adenylyl cyclase (through G_i_ α subunit) and the activation of phospholipase C-β (through G_i_ βγ subunits) [47,48,49,50].

### 2.3. Degradation of CNP

The plasma CNP concentration is considerably low [25,51]. This is because CNP is rapidly degraded in circulation or in the periphery where it acts as an autocrine/paracrine factor. The plasma half-life of CNP is reportedly 2.6 min [52]. The main catabolic pathways of CNP include its clearance receptor, NPR-C, referred to as the c-receptor, and neutrophil endopeptidase 24. 11 (NEP).

NPR-C is the third identified natriuretic peptide receptor, succeeding NPR-A and NPR-B. NPR-C has a similar affinity to all three natriuretic peptides; however, the binding affinity is as follows in both humans and rats: ANP > CNP > BNP [7]. Whereas NPR-A and NPR-B are biologically active receptor guanylyl cyclases with catalytic or enzymatic domains under the membranous portion, NPR-C does not have intracellular guanylyl cyclase domains and, thus, cannot produce cGMP as the second messenger. It internalizes the bound ligands and degrades them intracellularly [46]. The role of NPR-C in the metabolism of CNP is summarized elsewhere [53]. In addition, there exists an intrinsic and specific ligand for NPR-C named osteocrin or musclin. Osteocrin can regulate the effect of CNP by moderating the local CNP concentrations [54,55,56].

NEP is a zinc-dependent peptidase that is present in numerous tissues, including the lung, kidney, endothelial cells, and plasma. NEP degrades natriuretic peptide family members, and CNP is reportedly highly susceptible to degradation by NEP in vitro [57]. Later, NEP was shown to regulate CNP metabolism in in vivo infusion experiments [58], and its inhibition was exhibited to enhance CNP-related actions in several tissues [59,60,61].

### 2.4. Distribution of CNP

Whereas ANP and BNP are known as cardiac hormones which are produced in the atrium and ventricle of the heart, respectively, CNP is expressed ubiquitously throughout the body and is produced in various tissues. Together with the fact that the plasma concentrations of CNP are relatively low as mentioned above, CNP is thought to be an autocrine/paracrine factor. Suga et al. revealed that CNP is expressed in endothelial cells for the first time [18]. Since then, various studies have exhibited that endothelial cells express and secret CNP, predisposing that the endothelium is an important tissue in which CNP plays a pivotal role [62,63,64]. Generally, CNP is thought to have vasodilatory and antimitogenic actions there. Furthermore, Komatsu et al. first reported that CNP exists in the brain, including the pituitary gland [14], and many studies followed with the discovery of the production and the expression of CNP in the central nervous system [65,66,67,68,69,70,71], indicating important roles of CNP there.

As the most prominent phenotype of systemic CNP knockout mice was impaired skeletal growth [23], the physiological role of CNP in skeletal tissues, especially that in the growth plate cartilage, was presumed to be the most important among those in all tissues in the mammalian body. The notion that the CNP/NPR-B pathway is crucial for skeletal growth was confirmed in rat experimental models [72,73] and further in humans through the observation of several pathophysiological phenotypes in cases with genetic mutations in the genes coding for relevant molecules included in this pathway [74,75,76]. Using this prominent growth-promoting effect of the activation of the CNP/NPR-B pathway on skeletal tissues, a CNP analogue was developed to improve the impaired skeletal growth observed in patients with achondroplasia, one of the most common forms of skeletal dysplasia [31,32].

Another obvious phenotype of systemic CNP knockout mice is infertility. Concerning this point, the CNP/NPR-B system in the reproductive system was investigated, and its critical role in female fertility was elucidated using mutant mice with impaired CNP or NPR-B function [19,20].

## 3. Physiological Roles of CNP in the Heart

### 3.1. Distribution of CNP and NPR-B in the Heart

Although widely expressed throughout the body as mentioned above, CNP is also found in the heart [77]. However, the expression levels are much lower than those of ANP or BNP [78,79]. Researchers could not detect CNP in the heart of rats in the earliest studies using radioimmunoassay [12,14,80]. Later, in various species including cartilaginous fish [81], *Squalus acanthias* [82], and *Triakis scyllia* [83], CNP itself and its gene expression were detected in rat heart [84]. CNP and its receptor NPR-B were also detected in goat cardiomyocytes [85]. Likewise, researchers were unable to detect CNP in human hearts in early studies [86], but the augmentation of CNP production in the failing heart made it easy to evaluate the existence of CNP in the heart [87]; as discussed in a later section, in the failing heart, the expression of CNP and the plasma concentrations of CNP are increased [27,88,89,90,91].

The hearts of vertebrates are roughly composed of two types of cells: cardiomyocytes and their interstitial fibroblasts. As for cardiomyocytes, Wei et al. confirmed the presence of CNP within the cardiomyocytes by immunohistochemistry and radioimmunoassay in human subjects for the first time [77]. Soon after, the expression of NPR-B was detected in cardiomyocytes isolated from rat ventricle, but the cGMP genesis by CNP in the experimental preparation was reportedly low [92]. Later, CNP and NPR-B were detected in rat cardiomyocytes both in vitro and ex vivo [78].

On the other hand, CNP was shown to be synthesized in and secreted from cardiac fibroblasts in in vitro experiments using rat cultured ventricular cells. In an immunohistochemical study, NPR-B was detected at much greater levels in cardiac fibroblasts than in cardiomyocytes in frozen sections of rat ventricle. This was further confirmed by an immunoblot study using protein extracts of distinct cardiac cell types. Taken together, NPR-B in adult rat ventricle was reported to be predominantly confined to the nonmyocyte population [93]. In humans, NPR-B activity is increased in nonmyocytes in failing ventricles, possibly as a result of increased fibrosis, and human ventricular cardiomyocytes were reported to express much lower levels or possibly no NPR-B [94].

### 3.2. Downstream Signaling of CNP in the Cardiac Cells

As for the downstream signaling of CNP in the cardiac cells, the signaling pathway through cGKs is regarded as one of the main pathways of the CNP/NPR-B/cGMP signaling cascade, as reviewed elsewhere [95]. Briefly, cGKs activated by cGMP inhibit calcium (Ca) signaling and suppress the calcineurin nuclear factor of activated T cells (NFAT) pathway in cardiac myocytes [96]. A nonselective non-voltage-gated cation channel, L-type Ca channel [97], and the transient potential canonical 6 (TRPC6) [98] are phosphorylated by cGKs and are, thus, involved in attenuating Ca entry, while also inhibiting Ca/calmodulin-activated kinase II (CaMKII). CNP greatly increased the phosphorylation of phospholamban (PLN) a and increased that of troponin I (TnI) to some extent in a failing heart model [99]. Several studies have revealed the central roles of the regulators of G-protein signaling (RGSs) (Figure 2).

Among PDEs, PDE3 is dominant in cardiomyocytes [93]. In particular, the CNP-mediated increase in cGMP is generally regulated by PDE2; however, through inhibition experiments, PED3 was shown to be more functionally important than PDE2 [99]. On the other hand, CNP sensitizes cAMP-mediated signaling in the non-failing heart via the NPR-B-mediated increase in cGMP, which inhibits the cAMP-PDE activity of PDE3 [100].

Natriuretic peptides including CNP reportedly modulate the current of ATP-sensitive potassium (K) channel in cardiomyocytes from the ventricle [101], which may be relevant to the fact that they can increase intracellular cGMP levels through NPR-A or NPR-B.

### 3.3. Physiological Effects of CNP on the Heart

In a physiological situation, CNP expression levels in cardiomyocytes are much smaller than those of ANP and BNP [78], and, as was the case with the exhibition of its existence, early studies failed to prove any effects of CNP on the heart [92]. Although a gene-targeting approach is fascinating to reveal the physiological roles of a gene product, mice depleted with CNP in cardiomyocytes or fibroblasts showed no obvious changes in the contractility, structure, or fibrosis of the heart, supporting the previous idea that CNP plays a minimal role in the heart in healthy conditions [102]. Nevertheless, NPR-B activity was shown to represent a significant portion of the natriuretic peptide-dependent guanylyl cyclase activity in the normal heart [103]. Yoshizumi et al. showed that CNP stimulates Na-dependent Ca efflux from freshly isolated adult rat cardiomyocytes [104]. Using ex vivo preparations of rat papillary muscle, CNP was shown to exert a positive lusitropic effect, in that the putative mechanism involved a cGMP-dependent enhancement of the rate of relaxation with a slowly developing negative inotropic effect [105]. CNP caused a significant reduction in the amplitude of contraction of cultured neonatal rat beating cardiomyocytes [106]. In contrast to endothelin-1, CNP reduced the contractility of these cells and further induced apoptosis via the accumulation of cGMP [107,108,109]. By using the hypertrophic rabbit heart model, negative inotropic effects of CNP were shown to be attenuated in hypertrophied ventricular myocytes because of reduced cyclic GMP production [110]. A recent study using ventricular myocytes isolated from transgenic mice expressing the highly sensitive cytosolic cGMP biosensor exhibited that NPR-B is evenly distributed across the ventricular muscle membrane and produces far-reaching, diffusible cGMP signals, whereas NPR-A is exclusively found in T-tubules where it creates a microdomain with restricted cGMP diffusion locally confined by PDE2 [111]. A Norway group also used targeted cGMP biosensors in rats and showed that CNP increases cGMP production near TnI, as well as sarcoplasmic reticulum Ca ATPase (SERCA), indicating that CNP can promote lusitropic and negative inotropic actions [112]. As for the molecular signaling pathway of the contractile effects of CNP in the heart, cGKI was demonstrated to be a downstream target, and cGMP/cGKI-stimulated phosphorylation of Ser16-phosphorylated PLN and subsequent activation of SERCA pump appear to mediate the positive lusitropic responses to CNP [113,114].

Summing up the above concept of the effect of CNP on heart contraction, NPR-B stimulation by CNP increases cGMP, and its downstream signaling cascade eventually causes a positive lusitropic and negative inotropic action in the myocardium. These effects are not mimicked by NPR-A stimulation by BNP, despite a similar cGMP increase.

CNP was reported to decrease fibroblast proliferation and extracellular matrix production in a NPR-B-mediated cGMP-dependent manner, i.e., CNP produced by cardiac fibroblasts is proposed to play a role in inhibiting cardiac fibrosis as an autocrine/paracrine factor [115].

## 4. Effects of CNP on Heart Failure

As mentioned, ANP and BNP are cardiac hormones secreted from the atrium and ventricle of the heart, respectively, and their clinical roles in patients with heart failure are well established; both of them are markers of the severity of heart failure and are further used as drugs for heart failure in clinical settings. Unlike ANP and BNP, CNP is thought to be a ubiquitous autocrine/paracrine regulator, and its expression levels in the heart are much lower than those of ANP and BNP; thus, so the role of CNP in heart failure did not initially attract much attention. Nevertheless, the gene expression of CNP and the plasma levels of NTproCNP were reported to be increased in case of heart failure, as along with ANP and BNP [27,88,89,90,91]. CNP production was increased in the hearts of patients with chronic heart failure, and this increase was correlated with the severity of heart failure [89,116]. Furthermore, the severity of the disease correlated with plasma NTproCNP levels, predicting all-cause mortality and hospitalization in patients suffering from heart failure with preserved ejection fraction (HFpEF) [117]. As CNP is an autocrine/paracrine regulator, in order to clarify whether increased levels of plasma proCNP are caused by increased production of CNP in the heart, CNP levels in the coronary sinus were compared with those in the aortic root in subjects with heart failure [116]. As a result, CNP levels in the coronary sinus were significantly increased compared with those in systemic circulation, indicating that CNP production is augmented in the failing heart. Dickey et al. reported that CNP generated twice as much cGMP as ANP in the mouse model of pressure-overloaded heart failure. They supposed that, in this condition, NPR-A activity decreased whereas NPR-B activity was not changed, indicating that NPR-B accounts for the majority of the natriuretic peptide-dependent activity in the failed heart [103].

As is the case with non-failing hearts, CNP is reported to show negative inotropic and positive lusitropic responses in rat failing heart models [118]. Concerning the role of PDEs that mediate the effects of CNP on failing hearts, the Oslo University group reported that the increase in global cGMP by CNP was mainly regulated by PDE2, not PDE3, in left-ventricular muscle strips and ventricular cardiomyocytes in the failing hearts of Wistar rats, but the functional consequences were different from the changes in cGMP, i.e., PDE3 inhibition induced the CNP-mediated negative inotropic response and lusitropic response, whereas PDE2 inhibition desensitized the CNP-induced negative inotropic response, but not lusitropic response. The increase in cGMP necessarily coincides with the functional responses and, generally, the functional responses induced by CNP are intermediately regulated by PDEs [99,100]. Furthermore, CNP sensitizes cAMP-mediated signaling via NPR-B-mediated increase of cGMP, which inhibits the cAMP-PDE activity of PDE3 in failing hearts [100].

The same group reported the involvement of SERCA activity as one of the effectors of the downstream molecules of CNP/NPR-B signaling, which mediates the negative inotropic and positive lusitropic effects of CNP in the failing heart. CNP-induced PLN and TnI phosphorylation by cGK in concert mediate both negative inotropic and positive lusitropic effects in failing hearts [119].

In addition, during the early phases of pressure overload, NPR-B/cGMP/cGKI signaling activated by CNP in cardiomyocytes protects from myocyte stiffening caused via titin [120].

## 5. Roles of CNP on Cardiac Hypertrophy

Similar to the case of heart failure, ANP and BNP are also biomarkers for cardiac hypertrophy. At the early stage of research on CNP, the role of CNP in cardiac hypertrophy was investigated as one of the cGMP generators along with other natriuretic peptides. The first intensive study on the specific effect of CNP on cardiac hypertrophy was performed by Tokudome et al. [121]. They investigated the effects of CNP on cultured cardiac myocyte hypertrophy and the interaction between CNP and endothelin-1 (ET-1), which is a representative stimulator of cardiac hypertrophy. Resultantly, CNP attenuated basal and ET-1-induced hypertrophy-related gene expression and inhibited ET-1-induced cardiomyocyte hypertrophy via a cGMP-dependent mechanism. The same group further reported that CNP affected antihypertrophic action in rat myocardial infarction (MI) models [122]. In addition, CNP reportedly attenuated angiotensin II-induced cardiac hypertrophy, fibrosis, and contractile dysfunction, which were accompanied by reduced cardiac superoxide production, in in vivo experiments using mice models [123]. On the contrary, the effects of CNP on cardiac contractility, guanylyl cyclase activity, and phosphorylation of cGMP-dependent protein were dampened in myocytes from the hypertrophied heart of mice induced by aortic banding [124].

As for other model animals, NPR-B dominant-negative transgenic rats displayed progressive, blood pressure-independent cardiac hypertrophy, and the hypertrophic phenotype was further enhanced in chronic volume overload-induced congestive heart failure, suggesting the preventing effect of CNP/NPR-B on cardiac hypertrophy [125].

## 6. Roles of CNP on MI

The vasculature is the major tissue where CNP abundantly exists and on which CNP potently works. As for the physiological and pathophysiological roles of CNP in the vasculature, the vascular natriuretic peptide system is proposed, in that endothelium-derived CNP affects vascular smooth muscle cells expressing its cognate receptor, NPR-B, and regulates vascular tone, remodeling, and regeneration [63,126]. On the other hand, endothelium-derived CNP is reported to maintain vascular homeostasis through NPR-C [127], and, in the case of MI, the CNP/NPR-C signaling governs coronary blood flow and protects against ischemia/reperfusion injury (I/R) complicated by MI [128].

In vivo administration of CNP was shown to attenuate cardiac remodeling after MI through its antifibrotic and antihypertrophic action [122]. In a mouse model with targeted overexpression of CNP in cardiomyocytes, overexpressed CNP did not affect I/R-induced infarct size but prevented cardiac hypertrophy induced by MI [129]. On the other hand, in a swine model of induced MI with preserved left-ventricular ejection fraction, CNP expression was locally increased in the infarct-remodeled myocardium in the presence of a dense capillary network, and a high concentration of CNP was required in the vasculogenic response there together with VEGF-A [130].

During MI and I/R, cGMP triggers cytoprotective responses and improves cardiomyocyte survival; cGMP production leads to the activation of cGKI, which in turn phosphorylates many substrates and eventually facilitates the opening of mitochondrial ATP-sensitive K channels and Ca-activated K channels of the BK type [131,132]. PDEs and SERCA2, which were mentioned above as downstream molecules in the CNP/NPR-B signaling cascade in the failing heart, are thought to be effective in MI states because the in vitro experimental model of heart failure included ventricular cardiomyocytes from Wistar rats with heart failure after MI [99,119].

In clinical settings, although circulating signal peptides of CNP could not identify patients with MI or those with unstable angina, it was significantly lower in patients with a history of previous MI and could identify those at risk of death or reinfarction within 1 year [133]. Similarly, plasma NTproCNP is reported to be an independent predictor of mortality and cardiac readmission in individuals with unstable angina [134]. A recent study performed in Denmark hospitals showed that increased proCNP levels at admission are an independent risk of all-cause mortality in female patients with ST-elevated MI [135].

## 7. Effects of CNP on Heart Rate and Electrical Conduction in the Sinoatrial Node (SAN)

Chronotropic effects of CNP, i.e., effects of CNP on the heart conduction system, have also been reported. After indicating that CNP exerts a significant and prolonged positive chronotropic effect both in vivo and in vitro using a dog model [136], Beaulieu et al. showed CNP modifies cardiac ionic currents to produce positive chronotropic effects by stimulation of NPR-B located in the SAN region [137]. Subsequently, natriuretic peptides (including CNP) and their cognate receptors were shown to modulate ion channel function in the SAN [138,139,140]. Recently, to investigate the physiological roles of NPR-B signaling in regulating heart rate and SAN function, NPR-B-deficient mice were used, and it was revealed that NPR-B plays an essential physiological role in maintaining normal heart rate and SAN function by modulating ion channel function in SAN myocytes via a cGMP/PDE3/cAMP signaling mechanism [141].

## 8. Conclusions and Further Discussion

Despite fewer dynamic changes compared to ANP or BNP, CNP plays a distinct role in cardiac physiology and pathophysiology. Cardiac cell-specific manipulation of CNP elucidated its autocrine/paracrine mechanism of action on the heart; a recent study reported that CNP originating from cardiomyocytes, endothelial cells, and cardiac fibroblasts is essential in maintaining the cardiac structure, function, and coronary vasoreactivity [102]. In the physiological state, cardiomyocyte- and fibroblast-specific knockout mice showed no alteration compared to control mice regarding cardiac contractility, structure, or fibrosis, supporting the concept that CNP plays a minimal role in a healthy state. On the other hand, when artificial heart failure was induced by pressure overload after aortic banding, mice with specific depletion of CNP in cardiomyocytes and in fibroblasts both had decreased ejection fraction, increased ventricular dilation, and increased collagen deposition; in particular, in cardiomyocyte-specific knockout mice, cardiac hypertrophy was observed. Similar effects were observed in clinical settings; therefore, the development of drugs related to the cardioprotective effect of CNP-NPR-B signaling is expected. As an analogue for CNP was recently approved as a drug for impaired skeletal growth in achondroplasia, relevant remedies targeting cardiovascular disorders may be developed in the future. In addition, the development of a method for measuring CNP or NTproCNP is lacking compared to other natriuretic peptides. Authentic or standard measuring procedures for CNP or NTproCNP should be explored.

As for the receptor, NPR-B is regarded as biologically active because of its massive cGMP generation upon ligand CNP binding. However, NPR-C has come to attract attention as it can augment regional CNP content. Therefore, osteocrin, the intrinsic and specific ligand for NPR-C, may play a major role in cardiac physiology and pathophysiology (Figure 2). Using a wildtype or osteocrin knockout mouse model, Miyazaki et al. showed the possibility that osteocrin suppresses the worsening of chronic heart failure after MI by inhibiting the clearance of the natriuretic peptide family including CNP [55]. Szaroszyk et al. performed RNA sequencing on wasting murine skeletal muscles in the condition of heart failure and found a reduced osteocrin expression. Furthermore, by generating mice with skeletal muscle-targeted depletion or overexpression of osteocrin, they demonstrated that, under the pressure overload condition, the progression of cardiac dysfunction and myocardial fibrosis is adversely correlated with skeletal muscle osteocrin levels. As for the mechanism, osteocrin enhanced the abundance of CNP, which promoted cardiomyocyte contractility via protein kinase A and further inhibited fibroblast activity via cGK signaling. They also found that osteocrin expression was reduced in the skeletal muscle of patients with heart failure, suggesting the therapeutic potency of the augmentation of osteocrin for cardiac cachexia [142]. The activation of NPR-C by its cognate ligands osteocrin or its analogues may represent a novel therapeutic approach to various cardiac disorders.

## Figures and Tables

**Figure 1 biology-11-00911-f001:**
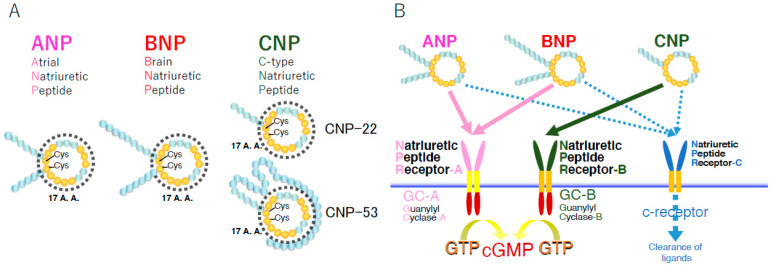
Schematic representation of natriuretic peptides (**A**) and their receptors (**B**). A.A.: amino acids; Cys: cysteines making the disulfide bond in each ring structure.

**Figure 2 biology-11-00911-f002:**
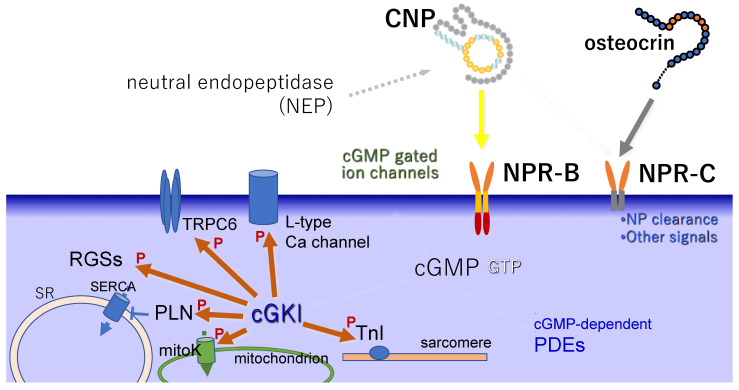
Schema of intracellular signaling molecules downstream of CNP/NPR-B. ‘P’ indicates the action of phosphorylation. SR: sarcoplasmic reticulum, SERCA: SR Ca-ATPase, PLN: phospholamban, RGS: regulator of G-protein signaling, TRPC6: transient potential canonical 6, TnI: troponin I, cGKI: cyclic GMP-dependent protein kinase I, mitoK: mitochondrial ATP-sensitive K channels and Ca-activated K channels of the BK type, PDE: phosphodiesterase.

## Data Availability

Not applicable.

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
