# Peer review of "Physiological and Pathophysiological Effects of C-Type Natriuretic Peptide on the Heart"

_biology, 2022, doi:10.3390/biology11060911_

Round 1

Reviewer 1 Report

The review is also interesting because at present the role of CNP is unclear. Unlike BNP and ANP the serum concentration is minimal so its role is seen as an autocrine/paracrine factor. Its production is ubiquitous and a role in reproduction has also been hypothesized.
Comments: the authors should cite a previous review on CNP specifying what is new on the topic after 2010. Curr Pharm Des. 2010; 16(37): 4080-4088. doi: 10.2174/138161210794519237 C-TYPE NATRIURETIC PEPTIDE (CNP): CARDIOVASCULAR ROLES AND POTENTIAL AS A THERAPEUTIC TARGET
Although the review is interested in cardiologic aspects it would be interesting to specify the role of CNP in pathologic situations in other tissues since its expression in cardiac fibroblasts suggests a concomitant role in fibroblasts in other tissues, In particular whether there are studies showing whether the 'cardio protective effect is associated with a positive or negative effect in other tissues in situations of heart failure   
The authors report that one of the stimulating factors is cytokines that are also produced throughout the body and not only in hearth
Specify whether any natriuretic effect of CNP has been reported and its pathogenetic mechanism explained
Explain how in in vivo studies the effect of CNP can be distinguished considering that the NPRC receptor is common to other natriuretic factors. 
Explain the role of NTproCNP, where it is produced and where it is degraded to CNP

Author Response

Thank you for your reviewing and instructive comments.

the authors should cite a previous review on CNP specifying what is new on the topic after 2010. Curr Pharm Des. 2010; 16(37): 4080-4088. doi: 10.2174/138161210794519237 C-TYPE NATRIURETIC PEPTIDE (CNP): CARDIOVASCULAR ROLES AND POTENTIAL AS A THERAPEUTIC TARGET

⇒ I cited this report and added as the reference 33.

Although the review is interested in cardiologic aspects it would be interesting to specify the role of CNP in pathologic situations in other tissues since its expression in cardiac fibroblasts suggests a concomitant role in fibroblasts in other tissues, In particular whether there are studies showing whether the 'cardio protective effect is associated with a positive or negative effect in other tissues in situations of heart failure

⇒ Regrettably, there exist no reports on the effect of fibroblast derived-CNP on tissues out of the heart in case of heart failure. That might be because circulating CNP levels are so small that it cannot affect other tissues out of the heart. CNP expression in fibroblasts in other tissues out of the heart may not be changed in the situation of heart failure because of the different CNP expression pattern in each tissue.

The authors report that one of the stimulating factors is cytokines that are also produced throughout the body and not only in hearth
Specify whether any natriuretic effect of CNP has been reported and its pathogenetic mechanism explained

⇒ Thank you for your comment. Although cytokines reportedly stimulate CNP production in vitro (Suga et al. 1993), the pathophysiological role or significance is not elucidated. Hama et al reported the increased CNP levels in patients with septic shock; in such cases, cytokine-induced CNP may play any pathophysiological roles, but the detailed mechanism is not studied.

Explain how in in vivo studies the effect of CNP can be distinguished considering that the NPRC receptor is common to other natriuretic factors. 

⇒ The effect of CNP through NPRC is closely studied on its stimulating effect of bone growth in murine knockout experiments: the growth stimulating effect of NPRC knockout mice is not observed in CNP knockout background (Kanai et al., 2017). Regrettably, it is not elucidated in in vivo studies in the cardiovascular field.

Explain the role of NTproCNP, where it is produced and where it is degraded to CNP

⇒ NTproCNP is thought to be a bio-inactive N-terminal part which is produced through the breakdown of preproCNP, so it is not thought to play any physiological or pathophysiological roles in vivo except being possibly utilized as a biomarker for CNP production.

Reviewer 2 Report

Although similar review articles have been published, this article systematically described the biological feature of CNP and its role on heart.

Compared with the published review articles, this paper is a little less innovative. 

I suggest adding some content, such as the current status and prospects of the clinical application of CNP. 

In addition, I am also interested in the current testing methods for CNP or NT-proCNP in basic research as well as in clinical applications.

A minor issue should be noted. Please unify the writing of NT-proCNP and NTproCNP.

Author Response

Although similar review articles have been published, this article systematically described the biological feature of CNP and its role on heart.

Compared with the published review articles, this paper is a little less innovative. 

I suggest adding some content, such as the current status and prospects of the clinical application of CNP. 

⇒ As I could not access the most resent status of the development of CNP related drugs in the cardiac field, I added a relevant statement in the conclusion section (written with blue highlight).

In addition, I am also interested in the current testing methods for CNP or NT-proCNP in basic research as well as in clinical applications.

⇒ Following the above insertion, I also added a brief comment regarding CNP and NTproCNP measurement (written in red characters).

A minor issue should be noted. Please unify the writing of NT-proCNP and NTproCNP.

⇒ I unified NT-proCNP into NTproCNP. Thank you for your comment.